**Data Availability Statement:** All relevant data are within the paper and its Supporting Information files.

# Clinicopathological and prognostic significance of caveolin-1 and ATG4C expression in the epithelial ovarian cancer

Yuyang Zeng[1]¤, Mengxi Chen[1], Sridha Ganesh[1], Shunze Hu[2]*, Honglei Chen[1,3]*

**1** Department of Pathology, School of Basic Medical Sciences, Wuhan University, Wuhan, Hubei Province, P. R. China, **2** Department of Pathology, Maternal and Child Health Hospital of Hubei, Wuhan, Hubei Province, P. R. China, **3** Department of Pathology, Zhongnan Hospital of Wuhan University, Wuhan, Hubei Province, P. R. China

¤ Current address: Department of Ophthalmology, Renmin Hospital of Wuhan University, Wuhan, Hubei Province, P. R. China
* hushunze925@icloud.com (SH); hl-chen@whu.edu.cn (HC)

## Abstract

### Objective

Altered expression of caveolin-1 (CAV1) and autophagy marker ATG4C is observed in various types of human cancers. However, the clinical significance of CAV1 and ATG4C expression in epithelial ovarian cancer (EOC) remains largely unknown. The present study aims to explore the clinicopathological value and prognostic significance of CAV1 and ATG4C expression in EOC.

### Methods

The expression pattern and prognostic value of CAV1 and ATG4C mRNA in EOC were analyzed using data from the Cancer Genome Atlas (TCGA) database (N = 373). In addition, immunohistochemistry analysis was performed to detect and assay the expression of CAV1 and ATG4C proteins in tissue microarray of EOC.

### Results

Based on TCGA data, Kaplan-Meier analysis indicated that patients with low CAV1 mRNA (p = 0.021) and high ATG4C mRNA (p = 0.018) expression had a significantly shorter overall survival (OS). Cox regression analysis demonstrated that the expression levels of CAV1 (p = 0.023) and ATG4C mRNA (p = 0.040) were independent prognostic factors for OS in EOC. In addition, the Concordance Index of the nomogram for OS prediction was 0.660. Immunohistochemical analysis showed the expression levels of stromal CAV1 and cancerous ATG4C proteins, and high expression of both CAV1 and ATG4C protein in the stroma were found to significantly correlate with the histologic subtypes of EOC, especially with serous subtype.

**Funding:** This work was supported by the Joint Grant of Health Commission of Hubei Province (WJ2018H0151) for Dr.Hu and the Innovation Project of Wuhan University Medical School (No. MS2016001) for DR. Zeng. The funders had no role in study design, data collection and analysis, decision to publish, or preparation of the manuscript.

**Competing interests:** he authors have declared that no competing interests exist.

## Conclusions

Decreased expression of CAV1 mRNA and increased expression of ATG4C mRNA in EOC can predict poor overall survival. The expression levels of CAV1 protein in stromal cells and ATG4C protein in cancer cells are significantly associated with histologic subtypes of EOC. These findings suggest that CAV1 and ATG4C serve as useful prognostic biomarkers and candidate therapeutic targets in EOC.

## Introduction

Ovarian cancer remains the fifth leading cause of cancer death in women and the first leading cause of death from gynecologic cancers[1]. An approximate 295,414 new cases and 184,799 deaths occurred in 2018[2]. Due to the difficulties to treat and the lack of early symptoms and/ or early diagnostic biomarkers, the 5-year survival rate for all stages of ovarian cancer is less than 50% [1, 3]. Despite recent advances in the diagnosis and treatment, there is limited improvement in survival rates of ovarian cancer over the past decade. Among all pathological types, epithelial ovarian cancer (EOC) accounts for the vast majority. EOC is also heterogenous disease that its clinical etiologies and somatic mutations vary with the histological subtypes, the latter mainly including serous, clear cell, endometrioid and mucinous carcinoma [4, 5]. Due to the significant differences between subtypes, it's necessary to take histology into consideration during biomarker studies on EOC.

Cancer tissue is composed of cancer cells and stromal cells. The cancerous epithelium and stroma contribute to form the tumor microenvironment and have been proved to be associated with tumor growth, angiogenesis, cell invasion, chemoresistance and cell metabolism in ovarian cancer[6]. Previously, biomarker studies in EOC are predominantly based on epithelial tumor components. However, the stroma-epithelial interactions play a vital role in the formation and progression of cancers as well as the patient prognosis [7]. Therefore, the stromal compartment deserves the promising resources of novel biomarkers and still wait for further research.

Caveolin-1(CAV1) belongs to a class of small scaffolding proteins (18–24 kDa), participating in the composition of membrane microdomains, caveolae [8]. Caveolae are typically found in endothelial and smooth muscle cells, and most of the epithelial tissues [9]. Further, CAV1 is involved in diverse cellular processes including signal transduction, cell cycle, proliferation, apoptosis, cancer cell invasion, migration and metastasis [10]. CAV1 shows a tumor-dependent and tissue-dependent role of tumors. In cancer cells, CAV1 has both positive and negative impacts on cancer development—relating to cell invasion and metastasis [11] as well as cancer repression [12]. In the stromal cells, absence or loss of stromal CAV1 immunostaining is a novel indicator of poor prognosis in various human cancers, such as breast [13, 14], gastric [15] and prostate carcinomas[16], as well as in metastatic melanoma [17]. Furthermore, previous studies have indicated that CAV1 functions as a tumor suppressor in ovarian carcinoma cells, and stromal CAV1 is lost in the majority of malignant and borderline ovarian carcinomas [18, 19]. However, the exact role of CAV1 in the carcinogenesis of EOC is yet to be fully elucidated.

Autophagy, a sort of cellular cannibalism whereby the redundant or damaged cell contents are removed by the lysosomal degradation pathway. Autophagy has recently been demonstrated to involve in ovarian carcinogenesis and has indicative values of cancer therapy [20]. During autophagy, autophagosomes, a kind of double membrane vacuoles, are produced to

entrap and degrade the cell components, and then fuse with lysosomes [21]. As an autophagy marker, LC3 protein is characteristically located on the membranes of autophagosomes [22]. ATG4C has enzyme function to cleaving the C-terminal amino acids of LC3 protein for lipid conjugation and dissociating LC3 from the autophagic vesicle surface for recycling [23]. Regulation of response amplitude by ATG4 protein is considered as a critical checkpoint for autophagy control [24]. ATG4C has proved to be the most widely expressed ATG4 isoforms in human tissues[25]. Among numerous targets implicated in autophagy response, ATG4C seems to have some unique capabilities in ovarian carcinogenesis [26].

However, the expression level and clinical value of ATG4C in EOC remain unclear. The precise mechanism of autophagy in EOC also remains to be determined. The potential role of CAV1 and ATG4C in the formation and progress of cancers deserve further exploration. In the present study, we investigated the expression pattern and prognostic value of CAV1 and ATG4C mRNAs in EOC patients using data from TCGA database, and evaluated the expression of CAV1 and ATG4C proteins in cancer cells and stromal cells, as well as their clinical significance in various histologic subtypes of EOC by immunohistochemistry.

## Materials and methods

### Bioinformatic analysis of CAV1 and ATG4C in EOC patients from TCGA-Ovarian cancer

CAV1 and ATG4C mRNA expression in patients with EOC were examined using data from TCGA-Ovarian Cancer (TCGA-OV) database. As the gene expression profiling and clinical information of patients were available, 373 EOC samples were selected from TCGA (https://portal.gdc.cancer.gov) and normalized, formatted, and organized for analysis of the association between CAV1/ATG4C mRNA expression and overall survival (OS). Samples with incomplete information were removed before analysis. Clinicopathological parameters of these patients, including age at diagnosis, race, histologic grade, clinical stage, anatomic subdivision, venous invasion status, lymphatic invasion status, presence of tumor residual disease, survival status, and OS time were obtained for survival-related comparison.

### Selection of patients and TMA construction

A total of 105 formalin-fixed, paraffin-embedded (FFPE) tissues, including 95 cases of EOCs, 6 cases of ovarian adenomas and 4 cases of normal ovarian tissues, were collected from the Department of Pathology, Zhongnan Hospital of Wuhan University. The patients with EOC were diagnosed in the period from 2013 to 2015. Two pathologists (Hu S and Chen H) reconfirmed the histopathologic features of these samples independently. All ovarian carcinoma samples had been classified according to the International Federation of Gynecology and Obstetrics (FIGO) criteria[27]. The histological subtype and grade were evaluated based on 2014 WHO classification criteria[28]. The study was approved by the Ethics Committees of Medical College, Wuhan University (No:JC2019-022). Informed consent was not obtained from all participants and/or their legal guardian/s, because the data were analyzed anonymously.

The EOC patients were composed of 95 women with a mean age of 50 (range = 18–82) years. Clinicopathological features, including age, histologic subtypes, histologic grades, tumor size (T), lymph node metastasis (N), distant metastasis (M), and FIGO stage were shown in Table 2. For 79 of the EOC patients, there was sufficient tissue for analysis of CAV1 and ATG4C immunostaining in cancer cells and stromal cells.

Hematoxylin and eosin-stained sections of all 105 cases ovarian lesion specimens were reviewed and the most representative areas were selected for tissue microarray (TMA) construction. Two TMA slides were constructed with a tissue manual arraying instrument, as described in our previous study[29]. Two cores (diameter—1.5 mm) were removed from the selected area of each donor FFPE specimen and precisely arrayed in a recipient paraffin block. Next, 4μm thick sections were consecutively incised from the recipient block and transferred to poly-L-lysine-coated glass slides. Hematoxylin and eosin staining was performed on TMA for the confirmation of tumor samples.

## Immunohistochemistry

Immunohistochemistry (IHC) was performed to detect the expression of CAV1 (rabbit anti-human polyclonal antibody, 1:150 dilution, sc-894, Santa Cruz, USA), and ATG4C (rabbit anti-human polyclonal antibody, 1:100 dilution; ab75056, Abcam, USA) proteins, according to manufacturer's instructions. HRP-conjugated second antibody and DAB kit (Dako, Agilent Technologies, USA) were used to visualize antibody binding. Immunostaining reactivity was observed by using light microscopy (Olympus BX-53 with CCD DP73). The standard positive control provided by the manufacturer served as a positive control, and the primary antibody was replaced with PBS in negative controls.

## Evaluation of immunohistochemical staining

The signals obtained from the labeling cells were detected via microscopy. The IHC results were evaluated by two pathologists (Hu S and Chen H), who were independent and blinded to the clinical parameters of the study. Expression of CAV1 and ATG4C proteins were evaluated by a semi-quantitative grading method in which the overall score, i.e. the intensity and density (ID) score, was according to the positive area and the staining intensity grade for each marker, the detail showed in our previous report[30]. The stromal cells were identified based on cell morphology in the microscopic high power field. The scores were independently evaluated by two researchers who had to reach an agreement. If divergences appeared, a third researcher participated in the evaluation to obtain the final score.

The total score ranged from 0 to 12, and the average values of each marker in cancer cells and stromal cells were taken as the cut-off values, where 6.0 was defined as the cutoff score for CAV1 in cancer cells (an ID score $\geqq 6.0$ defined high expression and ID score $< 6.0$ indicated low expression), 7.7 was defined as the cutoff score for ATG4C in cancer cells (an ID score $\geqq 7.7$ defined high expression and ID score $< 7.7$ indicated low expression), 6.5 was defined as the cutoff score for CAV1 in stromal cells (an ID score $\geqq 6.5$ defined high expression and ID score $< 6.5$ indicated low expression) and 3.7 was defined as the cutoff score for ATG4C in stromal cells (an ID score $\geqq 3.7$ defined high expression and ID score $< 3.7$ indicated low expression).

## Statistical analysis

Statistical software package IBM SPSS Statistics 22.0 (IBM, Chicago, IL) and R software (R 3.3.2) was applied for statistical analysis. X-tile 3.6.1 software (Yale University, New Haven, CT, USA) was used to determine the optimal cut-off values for CAV1 and ATG4C mRNA expression in TCGA data[31]. The significant difference between the Kaplan-Meier survival curves was assessed by the log-rank test. Cox regression (proportional hazards model) was applied for univariate and multivariate prognostic factor identifications. Nomograms were constructed based on the results of Cox multivariate analyses in terms of OS. By combined evaluation of the C-index and calibration, the performance of the established nomograms was

effectively measured. Statistical correlations of protein expression and clinicopathological parameters were assessed using the chi-square test or Fisher's exact tests. For binary categorical data, the phi coefficient, a measurement of the degree of association between two binary variables, was used to determine the association between two markers. P values of less than 0.05 were considered statistically significant. Cases with missing data items were also included in the analysis by categorizing them as "N/A".

## Results

### CAV1/ATG4C mRNA expression in EOC and overall survival

Based on survival data from TCGA, the optimal cut-off values for CAV1 and ATG4C mRNA expression were determined by X-tile software. The $\chi 2$ log-rank values of CAV1 and ATG4C were 4.863 and 5.427, respectively. Patients were divided into high expression group and low expression group for further analysis (CAV1$\leq$ 3.98 and >3.98, ATG4C$\leq$ 3.14 and >3.14) (Fig 1).

### Identification of prognostic factors in EOC

Based on the TCGA data, the independent prognostic value of CAV1/ATG4C mRNA expression and clinicopathologic factors in relation to OS was analyzed using univariate and multivariate analysis (Table 1). The univariate Cox regression model revealed that age, race, histologic grade, anatomic subdivision, venous invasion, residual tumor, clinical stage, expression of CAV1 and ATG4C mRNA were associated with prognosis of EOC patients in terms of OS ($P<0.05$). Multivariate analysis after adjustment indicated that the expressions of CAV1 and ATG4C mRNA were independent prognostic factors for OS in EOC patients ($P<0.05$). Concurrently, age, race, residual tumor, and clinical stage were also independent predictors regarding patient OS.

### Validation of the prognostic value of CAV1 and ATG4C in EOC based on nomograms

To further validate the prognostic value of CAV1 and ATG4C mRNA in EOC, nomograms were constructed based on age, race, residual tumor, clinical stage, CAV1 and ATG4C mRNA expression. CAV1 and ATG4C mRNA expression were identified as independent indicators in terms of OS via multivariate analyses. As shown in the calibration plots, there were excellent agreements between the prediction by nomogram and the actually observed probability of survival (3-, and 5-year OS) (Fig 3). The concordance index (C-index) of the nomogram for OS prediction was 0.660(95%CI, 0.618–0.703).

### Demographic and patient characteristics

Demographic and clinical characteristics of EOC patients were summarized in Table 2. The median age was 50 (18–82) years. The largest group of patients in the study was younger than 60 years of age (81.1%). According to the histologic subtype, 47(49.5%) patients were serous, 17(17.9%) were mucinous, 30(31.6%) were endometrioid and 1(1.0%) were clear cell EOC. In the aspect of histologic grade, 21(22.1%) patients were I, 28(29.5%) were II and 46(48.4%) were III. The distribution of FIGO stage was as follows: 47 (49.5%) patients in stage I, 24 (25.3%) in stage II, and 3 (3.2%) in stage III and 21(22.1%) in stage IV.

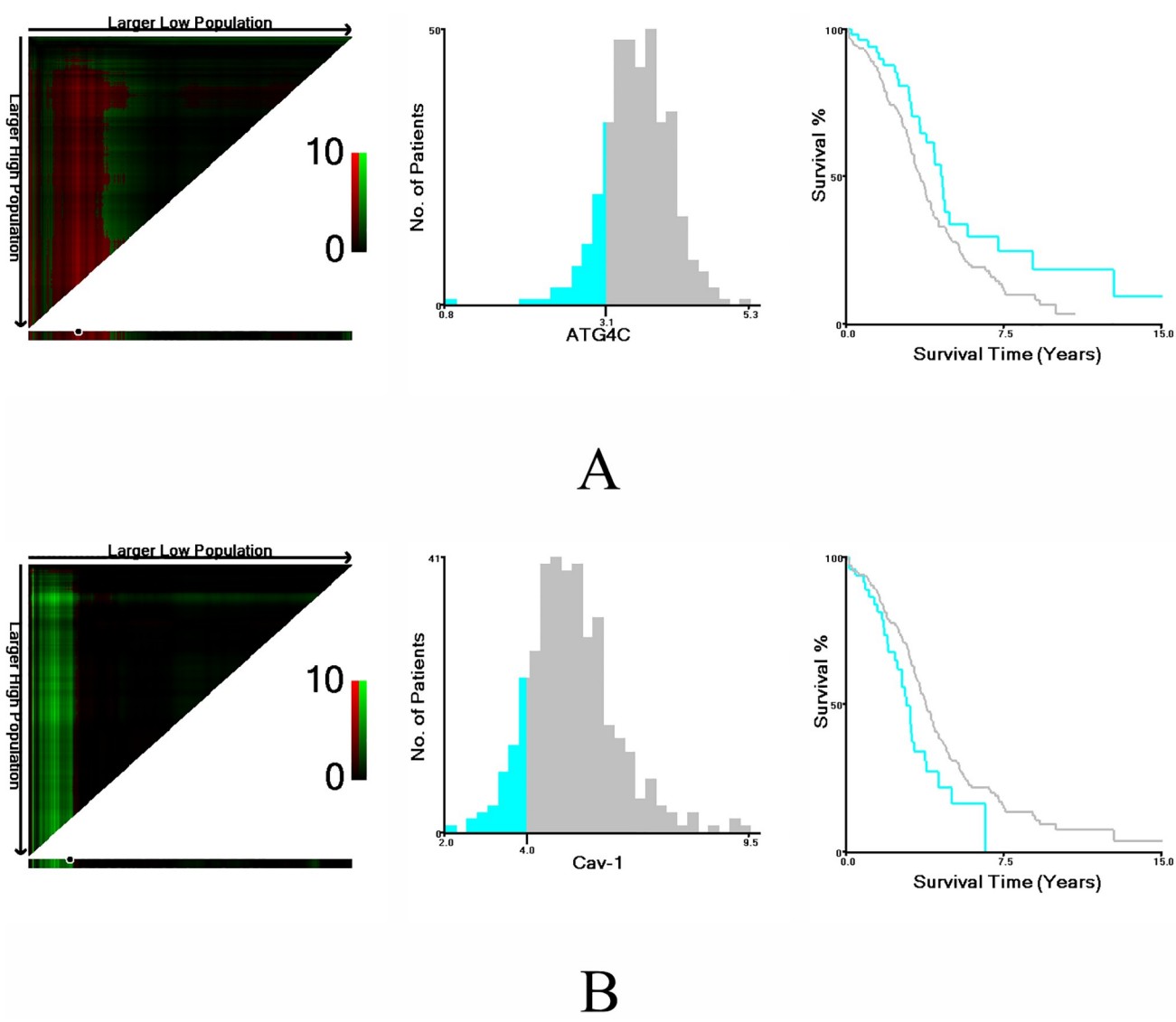

**Fig 1. Determination of cut-off values of ATG4C and CAV1 mRNA expression in TCGA database and survival analyses.** X-tile analysis of survival data in TCGA database was performed to determine the optimal cut-off value for ATG4C and CAV1 expression. The sample of ovarian cancer patients was equally divided into training and validation sets. X-tile plots of training sets are shown in the left panels, with plots of matched validation sets shown in the smaller inset. The optimal cut-off values highlighted by the black circles in left panels are shown in histograms of the entire cohort (middle panels), and Kaplan-Meier plots are displayed in right panels. P values were determined by using the cut-off values defined in training sets and applying them to validation sets. A) The optimal cut-off value for ATG4C was 3.14 ($\chi2 = 5.427$, P = 0.0201). B) The optimal cut-off value for CAV1 was 3.98 ($\chi2 = 4.863$, P = 0.0269). Subsequently, the differences in the Kaplan-Meier survival curves between patients with high and low expressions of CAV1 and ATG4C mRNA were assessed by the log-rank test. Results showed that the low-expression group of CAV1 mRNA (P = 0.021, Fig 2A) and the high-expression group of ATG4C mRNA(*P* = 0.018, Fig 2B) had significantly shorter OS.

## Expression of CAV1 and ATG4C in noncancerous and EOC tissues

All 95 EOC samples were available for analysis of CAV1 and ATG4C immunostaining in cancer cells, of which 79(83.2%) cases had sufficient tissues for analysis of CAV1 and ATG4C immunostaining in stromal cells. CAV1 was expressed in both EOC tissues and noncancerous ovarian tissues, predominantly localized in the cell membrane and cytoplasm (Fig 4). In noncancerous ovarian tissues, 10(100%) cases had high CAV1 expression in both epithelium and

**Table 1. Univariate and multivariate Cox proportional analysis with overall survival.**

| Variable | Univariate analysis | | Multivariate analysis | |
|---|---|---|---|---|
| | HR (95% CI) | P value | HR (95% CI) | P value |
| Age (years) | 1.019(1.006,1.032) | **0.005**\* | 1.023(1.009,1.037) | **0.001**\* |
| Race | | 0.070 | | **0.026**\* |
| Yellow | reference | | Reference | |
| White | 0.428(0.186,0.981) | **0.045**\* | 0.330(0.142,0.771) | **0.010**\* |
| Black | 0.673(0.261,1.739) | 0.413 | 0.479(0.177,1.292) | 0.146 |
| N/A | 0.308(0.076,1.241) | 0.098 | 0.185(0.045,0.761) | **0.019**\* |
| Histologic grade | | 0.059 | | |
| G1/G2 | reference | | | |
| G3/G4 | 1.389(0.911,2.119) | 0.127 | | |
| GX | 2.714(1.163,6.331) | **0.021**\* | | |
| Anatomic subdivision | | 0.134 | | |
| Right | reference | | | |
| Left | 0.520(0.297,0.912) | **0.022**\* | | |
| Bilateral | 0.777(0.500,1.209) | 0.264 | | |
| N/A | 0.773(0.368,1.627) | 0.498 | | |
| Venous invasion | | **0.006**\* | | |
| No | reference | | | |
| Yes | 0.782(0.414,1.474) | 0.448 | | |
| N/A | 1.519(0.922,2.504) | 0.101 | | |
| Lymphatic invasion | | 0.305 | | |
| No | reference | | | |
| Yes | 1.289(0.759,2.189) | 0.347 | | |
| N/A | 1.433(0.896,2.292) | 0.133 | | |
| Residual tumor | | **0.002**\* | | **0.010**\* |
| No macroscopic disease | reference | | reference | |
| 1–10 mm | 2.026(1.243.3.303) | **0.005**\* | 1.948(1.187,3.198) | **0.008**\* |
| 11–20 mm | 1.959(1.018,3.772) | **0.044**\* | 1.556(0.800,3.029) | 0.193 |
| >20 mm | 2.480(1.447,4.249) | **0.001**\* | 2.115(1.221,3.663) | **0.008**\* |
| N/A | 1.050(0.538,2.050) | 0.887 | 0.986(0.496,1.957) | 0.967 |
| Radiation therapy | | 0.211 | | |
| Yes | reference | | | |
| No | 0.300(0.027,3.335) | 0.327 | | |
| N/A | 0.189(0.026,1.368) | 0.099 | | |
| Clinical stage | | **0.043**\* | | **0.042**\* |
| Stage II | reference | | reference | |
| Stage III | 2.529(1.038,6.166) | **0.041**\* | 2.160(0.876,5.326) | 0.095 |
| Stage IV | 3.147(1.230,8.048) | **0.017**\* | 2.759(1.059,7.190) | **0.038**\* |
| N/A | 7.537(1.444,39,335) | **0.017**\* | 8.839(1.595,48.978) | **0.013**\* |
| CAV1 expression | | **0.022**\* | | **0.023**\* |
| Low vs. high | 0.630(0.423,0.937) | | 0.612(0.401,0.935) | |
| ATG4C expression | | **0.019**\* | | **0.040**\* |
| Low vs. high | 1.641(1.084,2.484) | | 1.557(1.021,2.373) | |

Abbreviations: G1: Well differentiated; G2: Moderately differentiated; G3: Poorly differentiated; G4: Undifferentiated; GX: Grade cannot be assessed.

\*indicates that the difference was statistically significant.

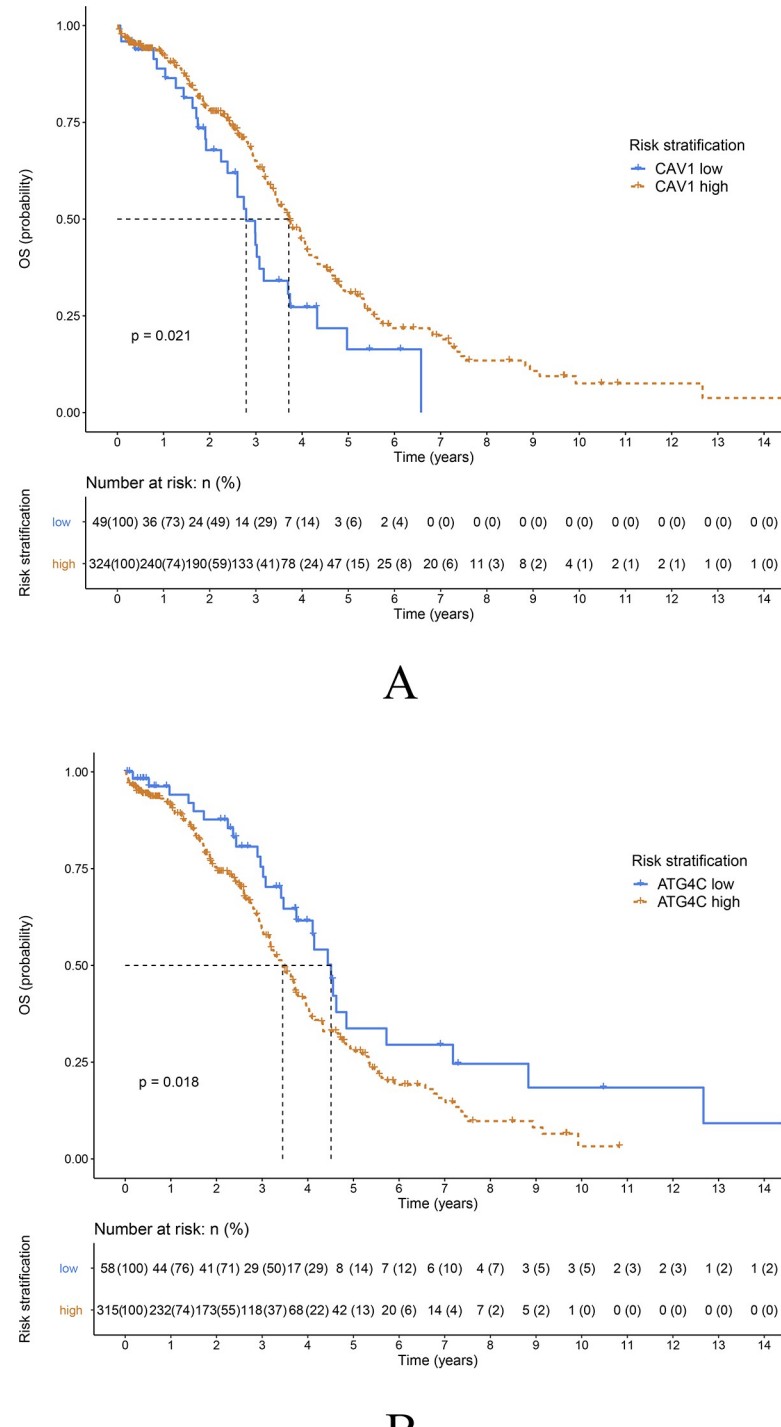

**Fig 2. Prognostic impacts of ATG4C and CAV1 mRNA expression in ovarian cancer in the TCGA database.** A)
Overall survival curves for patients with high or low CAV1 expression (P = 0.021, log-rank test). B) Overall survival
curves for patients with high or low ATG4C expression (P = 0.018, log-rank test).

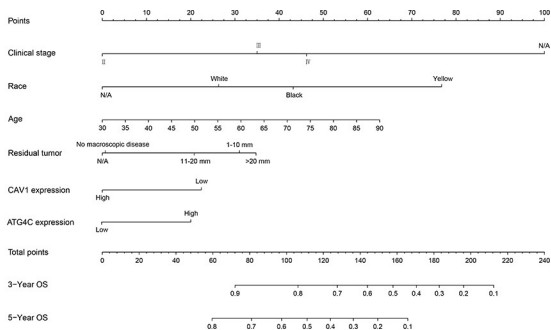

A

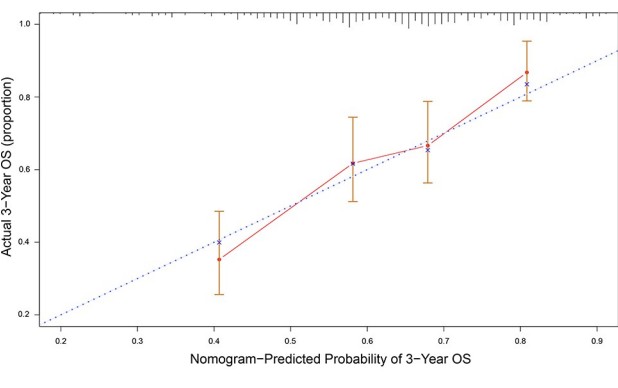

B

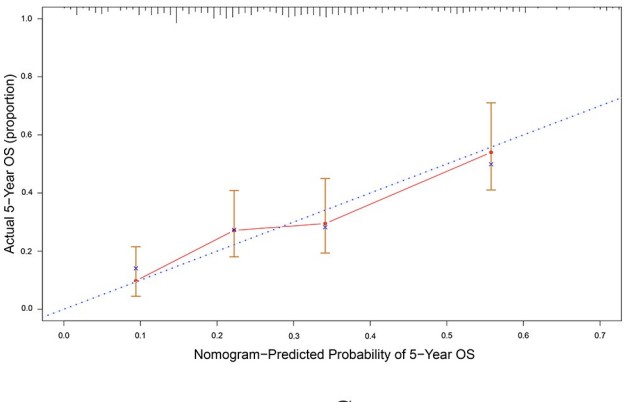

C

**Fig 3. Prediction model of ovarian cancer.** A) Prognostic nomogram for patients with ovarian cancer. The calibration curve of the nomogram for predicting OS at 3 years B) and 5 years C).

**Table 2. Patient characteristics (N = 95).**

| Characteristics | Sub-characteristics | Value (%) |
|---|---|---|
| Age (years) | | 50 (range = 18–82) |
| Histologic subtype | Serous | 47(49.5) |
| | Clear cell | 1(1.0) |
| | Mucinous | 17(17.9) |
| | Endometrioid | 30(31.6) |
| Histologic grade | I | 21(22.1) |
| | II | 28(29.5) |
| | III | 46(48.4) |
| Tumor size (T) | T1 | 47(49.5) |
| | T2 | 24(25.3) |
| | T3 | 24(25.3) |
| Lymph node metastasis (N) | N0 | 89(93.7) |
| | N1 | 6(6.3) |
| Distant metastasis (M) | M0 | 74(77.9) |
| | M1 | 21(22.1) |
| FIGO stage | I | 47(49.5) |
| | II | 24(25.3) |
| | III | 3(3.2) |
| | IV | 21(22.1) |
| Total | | 95(100) |

stromal cells, especially in the fibroblasts. However, in EOC tissues, there was negative CAV1 expression in inflammatory cells, but there was positive expression in vascular endothelial cells which acted as the positive internal control (Fig 4). High expression of CAV1 in cancer cells and stromal cells was 60.0% (57/95) and 50.6% (40/79) respectively, suggesting that CAV1 expression in EOC tissues was significantly reduced when compared with noncancerous ovarian tissues(P = 0.013, P = 0.002, respectively).

The association between CAV1 and ATG4C proteins was further investigated (Table 3). There was a positive association between CAV1 and ATG4C protein expression in cancer cells (P<0.001, phi coefficient = 0.368); a positive correlation between CAV1 and ATG4C protein expression in stromal cells was also identified (P = 0.002, phi coefficient = 0.345).

## Clinical significances of CAV1 and ATG4C protein expression

To investigate the effect of CAV1 and ATG4C expression on malignant progression, the correlation between protein expression and the clinicopathologic features, including age, histologic subtype, histologic grade, T, N, M, and FIGO stage of EOC, were examined respectively. As shown in Tables 4 and 5, high expression of CAV1 protein in the stroma and ATG4C protein in cancer cells were found to be significantly related to the histologic subtypes of EOC patients (P = 0.019, P = 0.005, respectively). Interestingly, ATG4C staining in cancer cells was strongly positive in the majority of serous EOC, which had poor prognosis, compared with the other histology subtypes. By analyzing the correlations between high or low expression of both CAV1 and ATG4 in the cancer (S1 Table) or stroma cells(S2 Table) and the clinicopathologic features of EOC patients, we additionally found that high expression of both CAV1 and

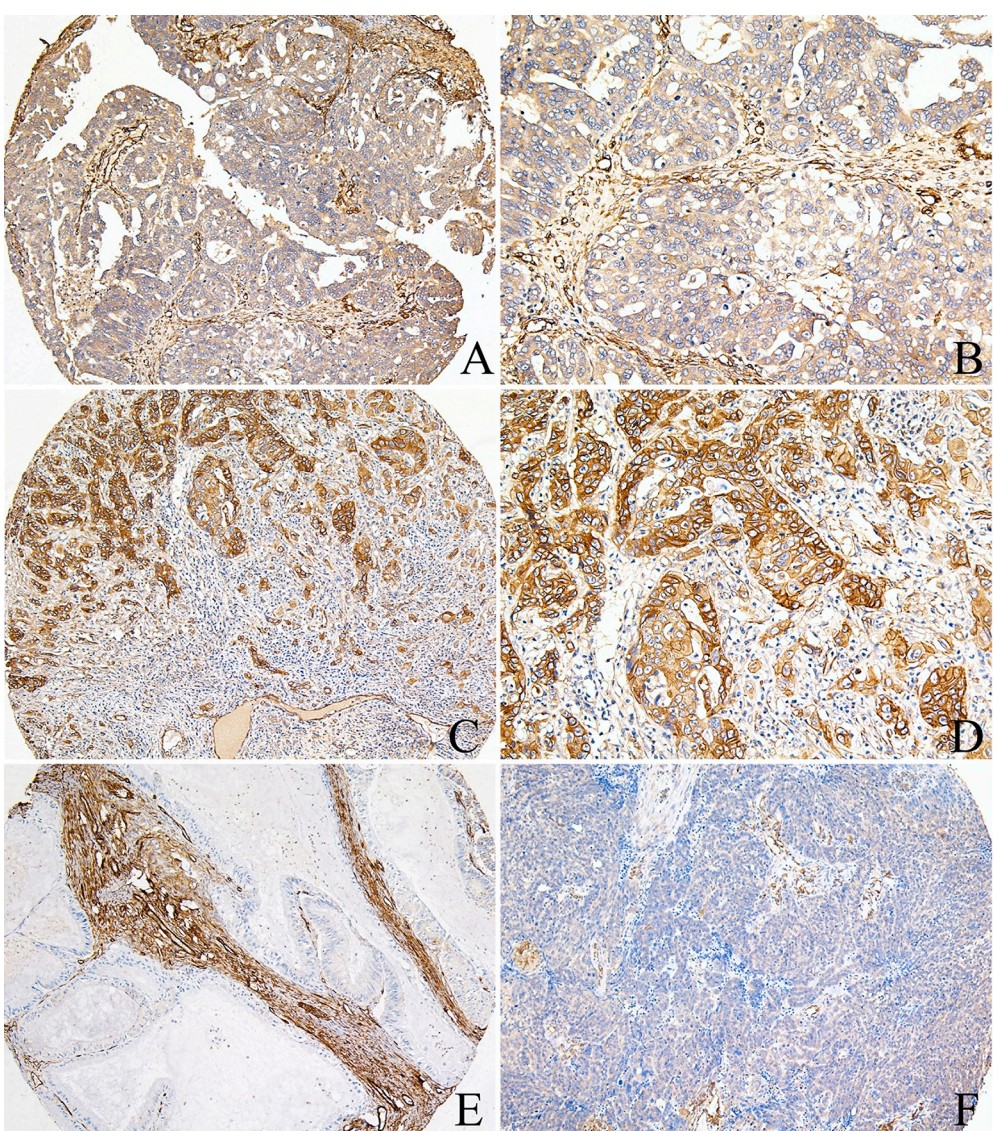

**Fig 4. CAV1 protein expression in ovarian cancer tissues.** A, B) High expression of CAV1 in the tumor cells and fibroblasts, positive in the vascular endothelial cells of ovarian serous cancer tissues; C, D) High expression of CAV1 in the cancer cells, negative in the inflammatory cells of ovarian endometrioid cancer tissues; E) Low expression of CAV1 in the cancer cells, high expression in the fibroblasts of ovarian mucinous cancer tissues; F) Low expression of CAV1 in the cancer cells and fibroblasts, negative in the inflammatory cells, positive in the vascular endothelial cells of ovarian endometrioid cancer tissues (A, C, E, F: 100×IHC; B, D: 200×IHC). ATG4C protein was highly expressed in EOC cells, primarily localized in the cell membrane and cytoplasm, moreover, there was positive ATG4C expression in cancer-related fibroblasts (CAFs) and inflammatory cells (Fig 5). Among EOC tissues, high expression of ATG4C in cancer cells and stromal cells was 70.5% (67/95) and 31.6% (25/79) respectively, which was higher than that in noncancerous ovarian tissues.

ATG4C protein in the stroma was significantly related to the histologic subtypes of EOC patients (P = 0.001).

## Discussion

Ovarian cancer has been a leading cause of death among women worldwide for the past decades. Tumorgenesis results from the mutational amplification of proto-oncogenes such as

**Table 3. Association between CAV1 and ATG4C protein expression in epithelial ovarian cancer tissues.**

| | ATG4C | | | | Fibroblastic ATG4C | | | |
|---|---|---|---|---|---|---|---|---|
| | Negative | Positive | *P* value | The phi coefficient | Negative | Positive | *P* value | The phi coefficient |
| CAV1 | | | **<0.001**\* | 0.368 | | | 0.370 | 0.101 |
| Negative | 19 (20.0) | 19 (20.0) | | | 23 (29.1) | 8 (10.1) | | |
| Positive | 9 (9.5) | 48 (50.5) | | | 31 (39.2) | 17 (21.5) | | |
| Fibroblastic CAV1 | | | 0.651 | -0.051 | | | **0.002**\* | 0.345 |
| Low | 9(11.4) | 30(38.0) | | | 33 (41.8) | 6 (7.6) | | |
| High | 11(13.9) | 29 (36.7) | | | 21 (26.6) | 19 (24.1) | | |

\*indicates that the difference was statistically significant.

**Table 4. Clinicopathologic features and distribution of CAV1 and ATG4C in cancer cells of 95 epithelial ovarian cancer patients.**

| Parameters | N | CAV1 | | | ATG4C | | |
|---|---|---|---|---|---|---|---|
| | | Low (%) | High (%) | *P* value | Low (%) | High (%) | *P* value |
| Age | | | | 0.521 | | | 0.690 |
| <60 years | 77 | 32(33.7) | 45(47.4) | | 22(23.2) | 55(57.9) | |
| ≧60 years | 18 | 6(6.3) | 12(12.6) | | 6(6.3) | 12(12.6) | |
| Histologic subtype | | | | 0.870 | | | **0.005**\* |
| Serous | 47 | 19(20.0) | 28(29.4) | | 7(7.4) | 40(42.1) | |
| Clear cell | 1 | 0(0.0) | 1(1.1) | | 0(0.0) | 1(1.1) | |
| Mucinous | 17 | 8(8.4) | 9(9.5) | | 6(6.3) | 11(11.5) | |
| Endometrioid | 30 | 11(11.6) | 19(20.0) | | 15(15.8) | 15(15.8) | |
| Histologic grade | | | | 0.708 | | | 0.934 |
| I | 21 | 10(10.5) | 11(11.6) | | 6(6.3) | 15(15.8) | |
| II | 28 | 11(11.6) | 17(17.9) | | 9(9.5) | 19(20.0) | |
| III | 46 | 17(17.9) | 29(30.5) | | 13(13.7) | 33(34.7) | |
| Tumor size (T) | | | | 0.185 | | | 0.817 |
| T1 | 47 | 20(21.2) | 27(28.4) | | 14(14.8) | 33(34.8) | |
| T2 | 24 | 6(6.3) | 18(18.9) | | 8(8.4) | 16(16.8) | |
| T3 | 24 | 12(12.6) | 12(12.6) | | 6(6.3) | 18(18.9) | |
| Lymph node metastasis (N) | | | | 0.680 | | | 1.000 |
| N0 | 89 | 35(36.8) | 54(56.8) | | 26(27.4) | 63(66.3) | |
| N1 | 6 | 3(3.2) | 3(3.2) | | 2(2.1) | 4(4.2) | |
| Distant metastasis (M) | | | | 0.189 | | | 0.519 |
| M0 | 74 | 27(28.4) | 47(49.5) | | 23(24.2) | 51(53.7) | |
| M1 | 21 | 11(11.6) | 10(10.5) | | 5(5.3) | 16(16.8) | |
| FIGO stage | | | | 0.185 | | | 0.817 |
| I | 47 | 20(21.1) | 27(28.5) | | 14(14.8) | 33(34.8) | |
| II | 24 | 6(6.3) | 18(18.9) | | 8(8.4) | 16(16.8) | |
| III/IV | 24 | 12(12.6) | 12(12.6) | | 6(6.3) | 18(18.9) | |

Data were expressed as count and percentage for categorical variables and analyzed by Chi-square test, Continuity correction, or Fisher's exact test.

\*indicates that the difference was statistically significant.

**Table 5. Clinicopathologic features and distribution of CAV1 and ATG4C in the stroma of 79 epithelial ovarian cancer patients.**

| Parameters | N | Fibroblastic CAV1 | | | Fibroblastic ATG4C | | |
|---|---|---|---|---|---|---|---|
| | | Low (%) | High (%) | P value | Low (%) | High (%) | P value |
| Age | | | | 0.420 | | | 1.000 |
| <60 years | 64 | 33(41.8) | 31(39.2) | | 44(55.7) | 20(25.3) | |
| ≧60 years | 15 | 6(7.6) | 9(11.4) | | 10(12.7) | 5(6.3) | |
| Histologic subtype | | | | **0.019*** | | | 0.065 |
| Serous | 42 | 17(21.5) | 25(31.6) | | 25(31.6) | 17(21.5) | |
| Mucinous | 14 | 5(6.3) | 9(11.4) | | 9(11.4) | 5(6.3) | |
| Endometrioid | 23 | 17(21.5) | 6(7.6) | | 20(25.3) | 3(3.8) | |
| Histologic grade | | | | 0.134 | | | 0.275 |
| I | 18 | 8(10.1) | 10(12.7) | | 10(12.7) | 8(10.1) | |
| II | 23 | 8(10.1) | 15(19.0) | | 15(19.0) | 8(10.1) | |
| III | 38 | 23(29.1) | 15(19.0) | | 29(36.7) | 9(11.4) | |
| Tumor size (T) | | | | 0.544 | | | 0.757 |
| T1 | 38 | 19(21.4) | 19(21.4) | | 27(34.2) | 11(13.9) | |
| T2 | 21 | 12(15.2) | 9(11.4) | | 13(16.5) | 8(10.1) | |
| T3 | 20 | 8(10.1) | 12(15.2) | | 14(17.7) | 6(7.6) | |
| Lymph node metastasis (N) | | | | 0.977 | | | 0.935 |
| N0 | 74 | 36(45.6) | 38(48.1) | | 50(63.3) | 24(30.4) | |
| N1 | 5 | 3(3.8) | 2(2.5) | | 4(5.1) | 1(1.3) | |
| Distant metastasis (M) | | | | 0.312 | | | 0.688 |
| M0 | 61 | 32(40.5) | 29(36.7) | | 41(51.9) | 20(25.3) | |
| M1 | 18 | 7(8.9) | 11(13.9) | | 13(16.5) | 5(6.3) | |
| FIGO stage | | | | 0.544 | | | 0.757 |
| I | 38 | 19(24.1) | 19(24.1) | | 27(34.2) | 11(13.9) | |
| II | 21 | 12(15.2) | 9(11.4) | | 13(16.5) | 8(10.1) | |
| III/IV | 20 | 8(10.1) | 12(15.2) | | 14(17.7) | 6(7.6) | |

Data were expressed as count and percentage for categorical variables and analyzed by Chi-square test, Continuity correction, or Fisher's exact test.

*indicates that the difference was statistically significant.

RAS and MYC, and the mutational inactivation of tumor suppressor genes such as p53, p16, and RB[32]. The multistage genomic events disturb the gene expression profile and bring about numerous genetic alterations in EOC patients, including CAV1 and autophagy-related genes. In this study, we assessed the expression of CAV1 and ATG4C and evaluated their prognostic values in EOC. Results demonstrated that low expression of CAV1 mRNA and high expression of ATG4C mRNA had significantly shorter OS. What's more, we first revealed that both CAV1 and ATG4C mRNA are independent prognostic biomarkers in EOC patients, which needs further prospective research. To further validate the prognostic value of CAV1 and ATG4C, nomograms were constructed for the first time based on CAV1/ATG4C mRNA expression, clinicopathological factors, and OS in the multivariate analyses. Our results also revealed that CAV1 protein level in the stroma, ATG4C protein level in cancer cells, and high expression of both CAV1 and ATG4C proteins in the stroma were related to histologic subtypes of EOC. The above results suggested that CAV1 and ATG4C held promise for serving as valuable prognostic factors of EOC patients.

Autophagy is a self-degradation mechanism by which senescent or redundant cellular contents are disposed to recycle energy for cellular homeostasis [33]. Previous studies have

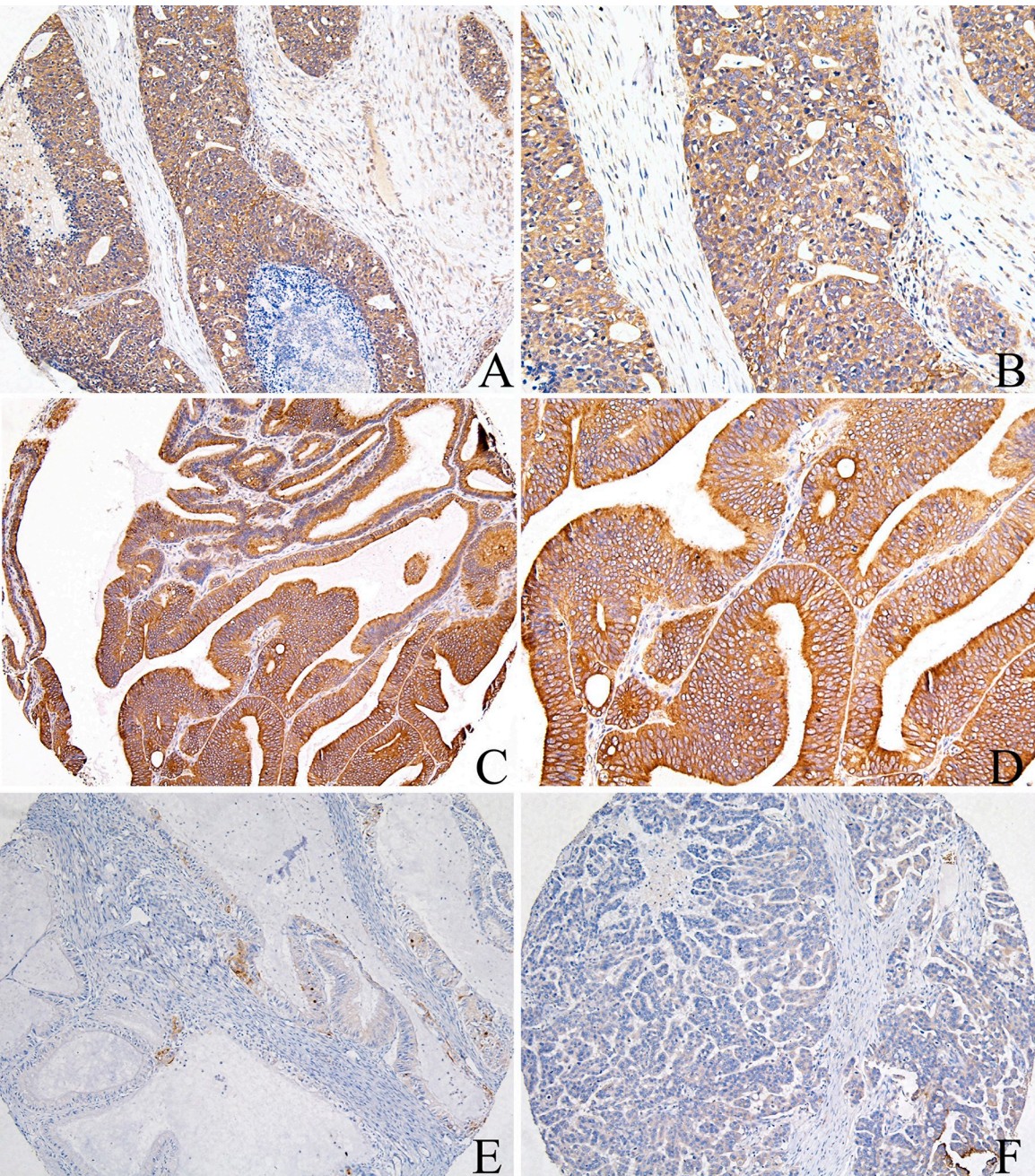

**Fig 5. ATG4C protein expression in ovarian cancer tissues.** A, B) High expression of ATG4C in the cancer cells, low expression in the fibroblasts of ovarian serous cancer tissues; C, D) High expression of ATG4C in the cancer cells of ovarian endometrioid cancer tissues; E) Low expression of ATG4C in the cancer cells, negative in the fibroblasts of ovarian mucinous cancer tissues; F) Low expression of ATG4C in the cancer cells, negative in the fibroblasts of ovarian endometrioid cancer tissues (A, C, E, F: 100×IHC; B, D: 200×IHC).

indicated that the activated autophagic flux in advanced human tumors usually correlates with malignant pathological phenotype and poor disease outcomes [34, 35]. ATG4C is a member of the cysteine proteinases family, which exerts the functions of the delipidation and deconjugation of the autophagy marker LC3 protein[36]. As an autophagy maker protein, ATG4C seems

to have some unique capabilities in ovarian carcinogenesis [24]. This study first demonstrated that high expression of ATG4C protein in EOC was significantly associated with histologic subtypes, especially serous subtype, and its mRNA predicted worse survival. Additionally, ATG4C mRNA was proven to be an independent prognostic marker in EOC patients and included in the nomogram for the first time.

As a main structural component of caveolae, CAV1 plays a crucial part in modulating cellular signaling. The previous studies have demonstrated that the decreased expression of stromal CAV1 promotes tumor aggressiveness in ovarian carcinoma[18] and low CAV1 mRNA expression in ovarian cancer tissues was associated with a worse prognosis[30, 37]. When compared with noncancerous ovarian tissues, our results showed that stromal CAV1 expression was lost in EOC tissues, which was consistent with the majority of evidence[38], and that CAV1 mRNA was an independent prognostic marker in EOC patients. Our results also demonstrated stromal CAV1 expression was significantly linked with the histologic subtypes of EOC. The tumor inhibiting role of stromal CAV1 can be explained in this way. The loss of stromal CAV1 has been reported to bring about activation of the tumor microenvironment, and induce the transformation of fibroblasts into CAFs [15]. Moreover, CAFs can influence the actual level of autophagy in ovarian cancer cells through the secretion of various pro-inflammatory cytokines and metabolites [39]. In addition, the activated tumor microenvironment is characterized by oxidative stress and hypoxia, which contribute to the stimulation of autophagy[40]. All of the above indicate that a metabolic cross-talk exists between stromal CAV1 and autophagy.

In this study, the results of CAV1 expression and clinical significances in EOC cells were a little different from the previous study. CAV1 is often expressed in advanced-stage ovarian carcinoma with metastasis [41]. Significant loss of CAV1 expression in EOC cells was observed, in comparison with noncancerous tissues which displayed 100.0% of high CAV1 expression among ten samples. And CAV1 expression in EOC cells was not significantly associated with histologic grade, distant metastasis, and FIGO stage. The cause of this difference may be due to discrepant sample sizes and different subtypes of ovarian cancers. However, the reduced expression level of CAV1 in EOC cells compared to noncancerous tissues may point to a role for CAV1 as a tumor suppressor gene[41]. Our study also showed a positive association between CAV1 and ATG4C protein expression in cancer cells and stromal cells separately. This may be explained as their dynamic synergic functions in carcinogenesis. Recent study indicated that stromal CAV1 favors tumor invasion and metastasis through force-dependent architectural regulation of the microenvironment. Moreover, both CAV1 and ATG4C are active participators in the process of cancer cell migration and invasion [42]. On the one hand, CAV1 regulates focal adhesion(FA) dynamics through tyrosine (Y14) phosphorylation, which stabilizes the FAK association with FAs, thus promoting cell migration and invasion[43]. On the other hand, FA dynamics play a key role in autophagy during cell migration and invasion [44]. Numerous abnormally large FAs accumulate in autophagy-deficient cancer cells, reflecting a role for autophagy in FA disassembly through targeted degradation of paxillin [45]. However, given that the high immunoreactivity of ATG4C and the abundant basal expression of CAV1 may cause a close correlation between the two, the exact underlying mechanism between CAV1 and ATG4C interactions in EOC is worthwhile for further in vitro study. Taken together, CAV1 and ATG4C may act as the targets or signaling molecules in cancer progression and contribute to cancer invasion and metastasis. There are limitations existing in the present study. Firstly, the sample sizes are insufficient for the algorithm of the stromal alteration in CAV1 and correlation analysis between CAV1 and ATG4C. Secondly, our study lacked deeper cell and tissue experiments, such as single cell sequence, Western blot and

polymerase chain reaction assays to verify the expression patterns and biological functions of CAV1/ATG4C in EOC.

In conclusion, this is the first study that evaluates the clinical significance and prognostic value of CAV1/ATG4C for EOC patients, as well as their expression and distribution in both cancer cells and stromal cells. Our data demonstrated that CAV1 and ATG4C mRNA is independently associated with the OS of EOC. Moreover, CAV1 protein level in the stroma, ATG4C protein level in cancer cells, and high expression of both CAV1 and ATG4C protein in the stroma are correlated with histologic subtypes of EOC patients. These findings suggest that CAV1 and ATG4C may be useful prognostic markers and potential therapeutic targets of EOC patients.

## Supporting information

**S1 Table. Clinicopathologic features of 95 epithelial ovarian cancer patients with high or low expression of CAV1 and ATG4C in cancer cells.**
(DOCX)

**S2 Table. Clinicopathologic features of 79 epithelial ovarian cancer patients with high or low expression of CAV1 and ATG4C in the stroma.**
(DOCX)

**S1 Data.**
(XLSX)

**S2 Data.**
(XLSX)

## Author Contributions

**Data curation:** Honglei Chen.

**Formal analysis:** Yuyang Zeng, Mengxi Chen, Shunze Hu.

**Funding acquisition:** Shunze Hu.

**Methodology:** Yuyang Zeng, Mengxi Chen, Sridha Ganesh.

**Project administration:** Honglei Chen.

**Software:** Yuyang Zeng.

**Supervision:** Shunze Hu, Honglei Chen.

**Writing – original draft:** Yuyang Zeng.

**Writing – review & editing:** Sridha Ganesh.

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
