## [Decision Letter · Decision Letter 0]

28 Jan 2020

PONE-D-20-00029

Clinicopathological and prognostic significance of caveolin-1 and ATG4C expression in the epithelial ovarian cancer

PLOS ONE

Dear Dr. Chen,

Thank you for submitting your manuscript to PLOS ONE. After careful consideration, we feel that it has merit but does not fully meet PLOS ONE’s publication criteria as it currently stands. Therefore, we invite you to submit a revised version of the manuscript that addresses the points raised during the review process.

The reviewer of your manuscript indicated that the work was preliminary and recommended that the manuscript be rejected. I am however willing to receive a substantially revised version of the manuscipt if you address comprehensively all the reviewers comments incorporating additional data as requested. Additional mechanistic insight is not necessary.

We would appreciate receiving your revised manuscript by Mar 13 2020 11:59PM. To enhance the reproducibility of your results, we recommend that if applicable you deposit your laboratory protocols in protocols.io, where a protocol can be assigned its own identifier (DOI) such that it can be cited independently in the future. For instructions see: http://journals.plos.org/plosone/s/submission-guidelines#loc-laboratory-protocols

We look forward to receiving your revised manuscript.

Kind regards,

Ivan R. Nabi, Ph.D.

Academic Editor

PLOS ONE

Journal Requirements:

2. We noticed minor instances of text overlap with the following previous publication(s), which need to be addressed:

(a) https://www.tandfonline.com/doi/full/10.4161/auto.8.2.18351

(b) https://www.spandidos-publications.com/10.3892/etm.2019.7402

(c) https://www.hindawi.com/journals/bmri/2014/462658/

(d) https://www.mdpi.com/2073-4409/8/7/713/htm

The text that needs to be addressed involves the Introduction section.

In your revision please ensure you cite all your sources (including your own works), and quote or rephrase any duplicated text outside the methods section. Further consideration is dependent on these concerns being addressed.

Reviewers' comments:

Reviewer's Responses to Questions

**Comments to the Author**

1. Is the manuscript technically sound, and do the data support the conclusions?

Reviewer #1: Partly

2. Has the statistical analysis been performed appropriately and rigorously? 

Reviewer #1: Yes

3. Have the authors made all data underlying the findings in their manuscript fully available?

Reviewer #1: Yes

4. Is the manuscript presented in an intelligible fashion and written in standard English?

Reviewer #1: Yes

5. Review Comments to the Author

Reviewer #1: The current manuscript by Dr. Chen and colleagues analyzed TCGA data and performed IHC on a TMA in order to determine whether Cav1 and/or ATG4C may correlate with ovarian cancer outcomes. In accordance with a previous study (Wiecher et al), they determined that reduced Cav1 levels associate with a poorer prognosis, particularly in serous cancers. They also determined that ATG4C positively associates with a poorer prognosis. These correlations appeared to occur at the level of mRNA and protein, and also were predictive even after multivariate analyses. Overall, while sound, these results are somewhat confirmatory and rather preliminary. While the TMA is a good resource, much of the analysis could be done online with k-plotter, for example. Hence, a great deal more data would be needed to generate a stand-alone publication.

Specific Comments

• The authors should expand the analysis to include additional sources of data (sequencing as well as tissue). They should also consider correlating Cav1 and ATG4C and confirming IHC findings by performing or analyzing stromal and cancer cells at a single cell resolution.

• The stromal alterations in Cav1 should be confirmed algorithmically in large patient data sets.

• Some mechanistic insight should be provided. How are these genes altered?

6. PLOS authors have the option to publish the peer review history of their article (what does this mean?). If published, this will include your full peer review and any attached files.

Reviewer #1: No

---

## [Author Response · Author response to Decision Letter 0]

4 Mar 2020

Dear Dr. Ivan R. Nabi,

Thank you very much for your letter and good advices on our manuscript entitled “Clinicopathological and prognostic significance of caveolin-1 and ATG4C expression in the epithelial ovarian cancer”, with the manuscript ID number PONE-D-20-00029. We have revised the manuscript, and will resubmit it for your consideration. We have carefully studied and addressed the comments raised by the academic editor and reviewer, and have made the corrections highlighted in red for our revised manuscript. Point by point responses to the reviewer’s comments are listed below this letter. 

We hope the revised manuscript is acceptable for publication in your journal. We look forward to hearing from you regarding our submission. And we would be glad to respond to any further questions and comments that you may have.

We would like to express our sincere thanks to the reviewers for the constructive and positive comments.

Honglei Chen

---

## [Editor Report · Decision Letter 1]

18 Mar 2020

PONE-D-20-00029R1

Clinicopathological and prognostic significance of caveolin-1 and ATG4C expression in the epithelial ovarian cancer

PLOS ONE

Dear Dr. Chen,

Thank you for submitting your manuscript to PLOS ONE. After careful consideration, we feel that it has merit but does not fully meet PLOS ONE’s publication criteria as it currently stands. Therefore, we invite you to submit a revised version of the manuscript that addresses the points raised during the review process.

The authors have addressed the majority of the comments raised previously. The presentation of a significant correlation between CAV1 and Atg4C in tumor and stromal cells in Table 3 is particularly interesting. It would be important therefore to present the clinicopathological features of not only high vs low CAV1 and high vs low ATG4C (Tables 4 and 5) but more particualrly whether the 48 high CAV1 and high ATG4 tumors or low CAV1/low ATG4 stroma present distinctive clinicopathological features.

We would appreciate receiving your revised manuscript by May 02 2020 11:59PM. To enhance the reproducibility of your results, we recommend that if applicable you deposit your laboratory protocols in protocols.io, where a protocol can be assigned its own identifier (DOI) such that it can be cited independently in the future. For instructions see: http://journals.plos.org/plosone/s/submission-guidelines#loc-laboratory-protocols

We look forward to receiving your revised manuscript.

Kind regards,

Ivan R. Nabi, Ph.D.

Academic Editor

PLOS ONE

---

## [Author Response · Author response to Decision Letter 1]

7 Apr 2020

Thank you very much for your letter and for the comments concerning our manuscript entitled “Clinicopathological and prognostic significance of caveolin-1 and ATG4C expression in the epithelial ovarian cancer” (The manuscript ID number: PONE-D-20-00029). Those comments are all valuable and very helpful for revising and improving our paper, as well as the important guiding significance to our researches. We have studied comments carefully and have made a necessary correction which we hope meet with approval. Revised portion is marked in red in the revised manuscript. The main corrections in the paper and the response to the reviewer’s comments are detailed below this letter.

---

## [Editor Report · Decision Letter 2]

10 Apr 2020

Clinicopathological and prognostic significance of caveolin-1 and ATG4C expression in the epithelial ovarian cancer

PONE-D-20-00029R2

Dear Dr. Chen,

We are pleased to inform you that your manuscript has been judged scientifically suitable for publication and will be formally accepted for publication once it complies with all outstanding technical requirements.

With kind regards,

Ivan R. Nabi, Ph.D.

Academic Editor

PLOS ONE
---

## [Editor Report · Acceptance letter]

1 May 2020

PONE-D-20-00029R2 

Clinicopathological and prognostic significance of caveolin-1 and ATG4C expression in the epithelial ovarian cancer 

Dear Dr. Chen:

I am pleased to inform you that your manuscript has been deemed suitable for publication in PLOS ONE. Congratulations! Your manuscript is now with our production department. 

With kind regards,

on behalf of

Dr. Ivan R. Nabi 

Academic Editor

PLOS ONE